# Vibration Performance Analysis and Multi-Objective Optimization Design of a Tractor Scissor Seat Suspension System

**Shuai Zhang [1], Weizhen Wei [1], Xiaoliang Chen [1,2], Liyou Xu [1,3,\*] and Yuntao Cao [4,5]**

[1] College of Vehicle and Traffic Engineering, Henan University of Science and Technology, Luoyang 471003, China

[2] School of Vehicle and Traffic Engineering, Henan Institute of Technology, Xinxiang 453000, China

[3] Advanced Manufacturing of Mechanical Equipment Henan Province Collaborative Innovation Center, Henan University of Science and Technology, Luoyang 471003, China

[4] General R&D Institute of China FAW Group Co., Ltd., Changchun 130022, China

[5] State Key Laboratory of Automotive Simulation and Control, Jilin University, Changchun 130022, China

[\*] Correspondence: xlyou@haust.edu.cn

**Abstract:** The combination of characteristic parameters is the key and difficult point to improving the vibration attenuation of scissor seat suspension. This paper proposes a multi-objective optimization method based on entropy weight gray correlation to optimize the combination of characteristic parameters with better vibration attenuation. The differential equation of seat suspension motion is derived through mechanical analysis, and a simplified driver seat suspension single degree of freedom model is constructed. The range of spring stiffness and damper damping is calculated theoretically. Through main effect analysis and analysis of contribution, the main influencing factors of seat suspension vibration attenuation are studied, and the influence correlation of the main factors is analyzed. On this basis, the spring stiffness and damper damping are taken as control variables, and the upper plane acceleration, displacement, and transfer rate of the seat suspension are taken as optimization objectives. The Optimal Latin Hypercube Sampling (OLHS) was used to sample the Design of Experiments (DoE), fit the RBF surrogate model, and screen the optimal solution based on the MNSGA-II algorithm and entropy weight gray relation ranking method. The comparative analysis of the performance before and after optimization shows that the vibration reduction performance response indexes of the acceleration, displacement, and transmissibility of the optimized seats are increased by 66.41%, 2.31%, and 8.19%, respectively. The design and optimization method proposed in this study has a significant effect on the vibration reduction of seats, which provides a reference for the optimization of the vibration reduction performance of seat suspension.

**Keywords:** scissor seat; suspension system; vibration performance; approximate model; multi-objective optimization

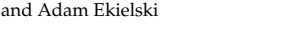



## 1. Introduction

Agricultural mechanization significantly impacts agricultural development, and its level is the main research focus on the agricultural machinery industry [1,2]. Tractors are an important piece of agricultural machinery, and they can be connected to various machines to perform a wide range of agricultural tasks [3]. Due to the strong vibration caused by the complex working conditions of the tractor [4], the driver's long-term work in this environment will impact his physical and mental health [5]. Tractors are bound to produce certain vibrations during farming operations, and the main reasons for these vibrations are the vibration of the whole vehicle caused by uneven road surfaces [6]; vibrations caused by sudden acceleration, braking, and steering of the vehicle while driving on the road [7]; the vibration caused by coupling when the engine and transmission

mechanism is rotating [8]; among them, the uneven road surface is the main cause of vehicle vibration. Therefore, it is meaningful to study the vertical vibration caused by uneven road surfaces [9]. In order to improve tractor ride comfort and driving safety, extensive research has been conducted to reduce tractor vibration from tractor tires, chassis suspension, engine mounting, and seats [10,11]. Improving tire and suspension parameters improves ride comfort while affecting tractor handling stability, braking performance, load-bearing capacity [12], etc. Comparatively speaking, because the seat system has the advantages of simple structure, convenient manufacture, and low cost, using seat suspension to improve tractor rides comfort is the most direct and effective method [13,14]. Tractors face complex operating conditions, and it is particularly important to improve seat suspension vibration attenuation [15].

In terms of seat suspension models, Desai R. et al. [12] validated that seat suspension with spring parallel and diagonal damper arrangement has better ride comfort than other arrangements, which provides guidance for seat suspension model design. Liao X. et al. [16] presented a mathematical model of a seat suspension system based on a negative stiffness structure, discussed the influence of different parameters on the dynamic stiffness of seat suspension, and obtained the ideal configuration parameters range of the suspension system. Du X.-M. et al. [17] introduced a simplified semi-active scissors seat suspension model to reduce the high amplitude vibration transmitted to the driver and proposed a robust state feedback H control based on an MR damper, which significantly reduced the vibration transmitted to the driver. For scissors-type seats with good stability and high reliability [18], they are widely used in agricultural tractors and commercial vehicles [19]. Therefore, the scissors-type seat is used as the research object in this study. In terms of seat suspension damping and stiffness control, Deng L. et al. [20] studied the influence of variable stiffness and damping on seat vibration attenuation, designed non-linear stiffness control and skyhook damping control, which significantly improved the vibration attenuation of seat suspension under different vibration excitations. Zhao Y. et al. [21] proposed a new fast system parameter identification method based on vibration test data for the seat system and estimated seat parameters such as stiffness and damping of the seat suspension system. Maciejewski I. et al. [22], constrained by seat suspension travel, designed an active finite-time sliding mode controller and implemented a multi-objective control strategy, which improved driver comfort with limited active power. He S et al. [23] took nonlinear damping coefficient and suspension stiffness as design variables and frequency-weighted RMS of driver's seat acceleration as the objective function and proposed an optimization method of ride comfort based on nonlinear damping and an intelligent algorithm, which effectively improves the practicability of seat model. In the tractor seat suspension, the spring stiffness and damper damping play a key role in affecting the vibration attenuation of the seat suspension [20–24]. Research on the combination matching of seat suspension spring stiffness and damper damping has become an outstanding problem to be solved urgently in order to improve seat vibration attenuation [25]. From the above study, stiffness and damping matching are key factors in improving the vibration attenuation of seat suspension.

Tractor seat suspension damping optimization covers both linear and non-linear factors, which makes the optimization of the seat suspension damping process consume a lot of human and material resources [26]. In order to improve the efficiency of optimization, researchers adopt the method of combining the agent model with an optimization algorithm to carry out a multi-objective optimization design [27]. The approximate model uses an efficient mathematical model or empirical formula to replace the actual analysis model without reducing the accuracy, thus greatly reducing the calculation cost of simulation cost and significantly improving the efficiency of optimization designs. At present, the commonly used approximate model methods include Response Surface Methodology (RSM), Radial Basis Function Neural Network (RBFNN), and Kriging [28]. Jiang R. et al. [29] took quality and fatigue life as optimization objectives, carried out a multi-objective optimization design for the control arm based on the Kriging proxy model and NSGA-II algorithm, determined

the optimal design of the control arm from Pareto solution by Entropy Weight Gray Relation Analysis (GRA), and proves the reliability and validity of the proposed multi-objective optimization method. Pandey M. et al. [30] used Radial Basis Function (RBF) integrated proxy modeling and NSGA-II algorithm to model the dynamic performance function of a truck with a three-piece bogie. The results show that the proposed optimization solution is significantly superior to the existing one.

Tractor seat vibration attenuation involves multi-disciplinary optimization designs of seat suspension structural dynamics, combination matching of characteristic parameters, structural safety, etc. Due to the interdependence and coupling among various disciplines, it is difficult to balance each performance using the traditional single-objective optimization method and to highlight the comprehensive vibration attenuation of seat suspension. Traditional multi-objective optimization methods mainly include a weighted average method, Benson method, interactive rule method, etc. The multi-objective optimization problem is transformed into a single-objective optimization problem through the weights between the objectives for solving, which is difficult to reflect the actual situation of the problem, and easy to cause local convergence and miss the optimal global solution [31]. Scholars have developed modern intelligent optimization algorithms based on population algorithms, mainly including improved Nondominated Sorting Genetic Algorithm (NSGA-II), Genetic Algorithm (GA), Particle Swarm Optimization (PSO) [32], etc. Modern intelligent optimization algorithm obtains Pareto solution by establishing a reasonable multi-dimensional mapping relationship between design space and target space, which more comprehensively reflects the actual situation of the problem, and improves the convergence accuracy and efficiency [33]. Prasad V. et al. [34] took cargo damage as a design criterion, proposed a GA-based fast convergence optimization algorithm, and compared the proposed algorithm with NSGA-II. This algorithm reduces calculation costs and performs better in cargo safety. Hua Y. et al. [35] aimed to stabilize suspension capability, used a multi-objective optimization method based on the RSM model, and proposed an improved Multiple Objective Particle Swarm Optimization (MOPSO) algorithm is proposed. The optimized design scheme has better vibration attenuation capability.

When considering vehicle use and research costs, the researchers focused on seat suspension in passenger and commercial vehicles [36,37]. Most tractor research enterprises neglect the consideration of driver's ride comfort, which lays a hidden danger to the driver's physical and mental health and driving safety [38]. Compared with heavy-duty engineering vehicles, tractors are required to meet more driving conditions due to the requirements of work tasks and carrying capacity, while the driving conditions of engineering vehicles are worse [39]. Therefore, the seat suspension of heavy-duty engineering vehicles needs to be designed for specific operating conditions. Relatively speaking, the design of tractor-seat suspension should meet the requirements of more driving conditions and lower costs. At present, the research on tractor-seat suspension mainly focuses on active and semi-active seat suspension. Vibration is attenuated by controlling stiffness and damping of seat suspension with a magnetorheological damper [40,41]. The purpose of this paper is to optimize the seat suspension system to improve the vibration attenuation according to the spring stiffness of the seat suspension and the damper damping range so that the tractor can have good vibration attenuation under most driving conditions. With the improvement of agricultural mechanization technology, people have higher requirements for the driving comfort of agricultural machinery. Therefore, as an important part of agricultural machinery, the demand for improving the comfort of tractor seats is increasingly significant.

To sum up, the vibration attenuation design of the seat suspension system mainly focuses on stiffness and damping control, and there is little research and application on tractors. Since the research mainly focuses on the active and semi-active control of the stiffness and damping of the seat suspension system, the high cost limits its application in agricultural machinery. The development of seat suspension based on the optimization of stiffness, damping, and structure design is more suitable for the application of tractors.

Therefore, this paper combines the approximate model method and the improved multi-objective optimization algorithm to optimize the tractor scissors seat suspension system.

In this paper, a scissors-type seat suspension is taken as a model, and a seat suspension mechanical model and a three-dimensional Adams model are established to analyze its dynamic performance. To study the influence law of seat suspension vibration performance and identify the main influencing factors of seat suspension vibration reduction. The multi-objective optimization method is used to optimize the vibration attenuation of seat suspension. The Pareto solution is sorted by the GRA method of Entropy Weight, and the optimal solution is selected. The vibration attenuation of the optimized before and after scissors seat suspension is compared and analyzed.

## 2. Methods

### 2.1. Main Effect Analysis

The main effect of a factor on the response is the average of the response of a factor to all tests at a certain level, so changing the level of a single factor, using the average of the effects of all possible combinations of each level and other factors on the results, provides the main effect. Typically, experimental designs calculate the main effect of a factor of response by constructing a multiple quadratic regression model based on the results of input factor and output response samples.

$$\bar{y}(x) = \bar{y}(x_1, x_2, \cdots, x_m) = \beta_0 + \sum_{i=1}^{m} \beta_i x_i + \sum_{i=1}^{m} \beta_{ii} x_i^2 + \sum_{i=1}^{m-1} \sum_{i<j=2}^{m} \beta_{ij} x_i x_j \tag{1}$$

As an example, the polynomial composition of the model with two input variables is as follows:

$$y = c_0 + c_1 x_1 + c_2 x_2 + c_3 x_1^2 + c_4 x_2^2 + c_5 x_1 x_2 \tag{2}$$

Derivation of the above second order polynomial yields:

$$dy = c_1 dx_1 + c_2 dx_2 + 2c_3 x_1 dx_1 + 2c_4 x_2 dx_2 + c_5 dx_1 x_2 \tag{3}$$

Then, $x_1$, $x_2$ the Linear Main Effect of the linear term is:

$$M_{x_1} = c_1 dx_1, M_{x_2} = c_2 dx_2 \tag{4}$$

$x_1$, $x_2$ the Linear Main Effect of the second order term is:

$$M_{x_1^2} = 2c_3 x_1 dx_1, M_{x_2^2} = 2c_4 x_2 dx_2 \tag{5}$$

Among them, $M_{x_1^2} = 2c_3 x_1 dx_1, M_{x_2^2} = 2c_4 x_2 dx_2$.
$x_1$, $x_2$ the interaction effect is.

$$M_{x_1 x_2} = c_5 dx_1 x_2 = c_5 (x_1 dx_2 + x_2 dx_1) \tag{6}$$

Among them,

$$d(x_1 x_2) = [Max(x_1)Min(x_2) + Min(x_1)Max(x_2)] - [Max(x_1)Max(x_2) + Min(x_1)Min(x_2)]$$

The main effect has the following laws: (1) numerical value (absolute value): the larger the value of the main effect, the greater the degree of influence of the factor on the response. (2) Direction: a positive main effect indicates that the response increases with the increase of the factor; conversely, a negative main effect indicates that the response becomes smaller with the increase of the factor. (3) Order: the presence of a second-order main effect means that the effect of the factor of the response is not linear. (4) Interaction effect: a large interaction effect means that two parameters varying simultaneously will greatly affect the response.

### 2.2. Analysis of Contribution

Contribution analysis mainly uses the Regression of DOE to calculate the contribution. According to the ranking of the contribution of design variables to performance objectives, design variables in the high discrete or high non-linear analysis are screened to reduce calculation costs and improve the efficiency of optimization design. The analysis and calculation steps are as follows.

Step 1: Normalization process

The DOE method is used to obtain experimental samples between design variables and performance objectives. The design variables have different design spaces, and the contribution values vary from the design space, requiring normalization of the sample data inputs using Equation (7).

$$x_i^* = \frac{x_i - \overline{x}}{\sigma} = \left( x_i - \frac{1}{N} \sum_{i=1}^{N} x_i \right) \times \left[ \sqrt{\frac{1}{N} \sum_{i=1}^{N} (x_i - \overline{x})^2} \right]^{-1} \tag{7}$$

where: $\overline{x}$ is the mean of the sample data, $\sigma$ is the standard deviation, $N$ is the total number of sample data, $x_i$ denotes the original input, and $x_i^*$ denotes the normalized input.

Step 2: Contribution Analysis

If there are $k$ design variables $(x_1, x_2, \ldots, x_k)$, then any performance objective can be formulated in a multiple regression model as:

$$P(x_1, x_2, \ldots, x_k) = \mu + \sum_{i=1}^{k} Q_i(x_i) + \ldots + \sum_{i=2}^{k} \sum_{j=1}^{k-1} R_{ij}(x_i, x_j) + \varepsilon \tag{8}$$

where: $P(x_1, x_2, \ldots, x_k)$ is any performance target, $\sum_{i=1}^{k} Q_i(x_i)$ is the main effect of the design variable, $\sum_{i=2}^{k} \sum_{j=1}^{k-1} R_{ij}(x_i, x_j)$ is the cross effect of any two design variables, $\mu$ is the total mean, $\mu$ is a constant term, and $\varepsilon$ is the error.

The main effects of the design variables can be expressed in Equation (9).

$$\sum_{i=1}^{k} Q_i(x_i) = \sum_{i=1}^{k} \hat{\beta}_i x_i \tag{9}$$

Therefore, the contribution values of the design variables can be defined by Equation (10).

$$N_{x_i} = \frac{100\hat{\beta}_i}{\sum_i |\hat{\beta}_i|} i = 1, 2, \ldots, k \tag{10}$$

where: $\hat{\beta}$ is the main effect coefficient $x_i$ of the design variable calculated by the least squares method; $N_{x_i}$ is the contribution of the design variable $x_i$.

### 2.3. Gray Relation Analysis

Gray Relation Analysis (GRA) is a method to measure the degree of approximation between experimental sequences and ideal sequences using Gray Relation Grade (GRG), which is widely used in multi-objective and multi-decision optimization problems and has comprehensive advantages in solving complex decision problems. The specific steps of GRA are as follows.

Step 1: Data pre-processing

Due to the different orders for magnitude of the experimental data, GRA may not be able to obtain a reliable optimized solution, and the experimental data need to be transformed into a set of dimensionless data $x$ onto $(0 < x < 1)$ for further quantitative

analysis. Depending on the characteristics of the response characteristics, different data preprocessing techniques can be used.

If the target has the characteristic of "bigger is better". The normalization method can be expressed as follows:

$$x_i^*(k) = \frac{x_i(k) - \min_k x_i(k)}{\max_k x_i(k) - \min_k x_i(k)} \tag{11}$$

If the objective is of a "lower is better" nature. The normalization method can be expressed as follows:

$$x_i^*(k) = \frac{\max_k x_i(k) - x_i(k)}{\max_k x_i(k) - \min_k x_i(k)} \tag{12}$$

If a specific value is ideal for the target. The normalization method can be expressed as:

$$x_i^*(k) = 1 - \frac{|x_i(k) - T|}{\max[\max_k x_i(k) - T, T - \min_k x_i(k)]} \tag{13}$$

where: $x_i^*(k)$ is the $k$th response characteristic value of the $i$th experiment after normalization; $x_i(k)$ is the initial design value of the response characteristic; $\min_k x_i(k)$ and $\max_k x_i(k)$ are the minimum and maximum values of all response characteristics $x_i(k)$, $k = 1, 2, \ldots n$, $i = 1, 2, \ldots m$; $m$ is the number of experiments; $n$ is the number of response characteristics; and $T$ is the specific value.

Step 2: Calculate Gray Relative Correlates (GRC)

The gray correlation coefficients are obtained. The gray correlation coefficient of the $k$th response characteristic of the $i$th experiment is expressed as:

$$\gamma(x_0^*(k), x_i^*(k)) = \frac{\triangle_{\min} + \zeta \triangle_{\max}}{\triangle_{0i}(k) + \zeta \triangle_{\max}} \tag{14}$$

where: $x_0^*(k)$ is the reference experimental sequence; $x_i^*(k)$ is the initial experimental sequence; $\triangle_{0i}(k) = |x_0^*(k) - x_i^*(k)|$ is the absolute difference between $x_0^*(k)$ and $x_i^*(k)$; $\triangle_{\max} = \max_i \max_k \triangle_{0i}(k)$ and $\triangle_{\min} = \min_i \min_k \triangle_{0i}(k)$ are the maximum and minimum values of $\triangle_{0i}(k)$, respectively; $\zeta$ is the identification coefficient, $\zeta \in [0, 1]$, generally defined as 0.5.

Step 3: Gray Correlation Gauge (GRG)

The gray correlation coefficient (GRC) was averaged to calculate the gray correlation degree (GRG), expressed as:

$$\Gamma(x_0^*, x_i^*) = \frac{1}{n} \sum_{k=1}^n \gamma(x_0^*(k), x_i^*(k)) \tag{15}$$

where: $\Gamma$ is the gray correlation degree, and $n$ is the number of response characteristics.

The relative significance of each response characteristic may be different, and the simple averaging method of Equation (15) may lead to inaccurate gray correlations (GRG). Therefore, the weighted gray correlation measure (GRG) is often calculated by assigning different weights to the response characteristics by Equation (16).

$$\Gamma(x_0^*, x_i^*) = \sum_{k=1}^n \omega_k \gamma(x_0^*(k), x_i^*(k)) \tag{16}$$

where, $\omega_k$ is the weight of the $k$th response characteristic, $\sum_{k=1}^n \omega_k = 1$.

### 2.4. Entropy Method

The Entropy Method is an objective assignment method for obtaining weight coefficients and is widely used in various fields [42]. "Entropy" was originally a physical thermodynamic concept but was later developed by C E Shannon into the entropy theory of information theory, and the "entropy" defined by C E Shannon is called "information entropy." Entropy is a measure of uncertain information. The smaller the amount of infor-

mation, the greater the uncertainty, the larger the entropy accordingly, and the smaller the weight in calculating the comprehensive evaluation value, while the larger the amount of information, the smaller the uncertainty, the smaller the entropy accordingly, and the larger the weight in calculating the comprehensive evaluation value.

The specific steps of the entropy method are as follows.

Step 1: dimensionless processing

In order to eliminate the influence of the difference in magnitude and order of magnitude between the initial performance indicators on the results, the initial performance indicators need to be dimensionless, consistent with the normalization method in the previous section. The evaluation matrix $P$ is:

$$p = \left(p_{ij}\right)_{m \times n} = \begin{pmatrix} p_{11} & p_{12} & \dots & p_{1n} \\ p_{21} & p_{22} & \dots & p_{2n} \\ \vdots & \vdots & & \vdots \\ p_{m1} & p_{m2} & \dots & p_{mn} \end{pmatrix} \tag{17}$$

where, $p_{ij}$ is the initial data after the normalization process.

Step 2: Calculation of index information entropy

According to the definition of information entropy, the indicator information entropy of $E_K$ is:

$$E_k = -H \sum_{i=1}^{m} f_{ki} \ln(f_{ki}) \tag{18}$$

where, $H = 1/\ln(m), f_{ki} = p_{ki} / \sum_{i=1}^{m} p_{ki}$, when $f_{ki} = 0$, let $f_{ki} \ln(f_{ki}) = 0$. $m$ is the number of evaluation objects, $f_{ki}$ is the $k$th performance index under the $i$th evaluation object accounted for the proportion of the performance index.

Step 3: Calculation of the entropy weight of the index

The information entropy of each indicator is calculated from the indicator information entropy formula $E_1, E_2, \dots, E_n$, then the entropy weight of the $k$th performance indicator is:

$$W_k = \frac{1 - E_k}{n - \sum_{k=1}^{n} E_k} \tag{19}$$

In the above equation, $n$ is the number of evaluation indicators, $0 \leq W_k \leq 1$ and, $\sum_{k=1}^{n} W_k = 1$.

### 2.5. RBF Approximation Model

The Radial Basis Function (RBF) is a typical pre-feedback control algorithm, which mainly contains an input layer, an implicit layer, and an output layer, and its structure is shown in Figure 1. Among them, the input layer is mainly used to introduce variable information, and the number of variables determines the number of dimensions of the input layer. The implicit layer is the core component of the RBFNN algorithm, which maps the input information on the output layer according to certain mathematical relationships to a built-in nonlinear algorithm. The number of neurons in the implicit layer is directly related to the learning ability, fault tolerance, and approximation accuracy of RBF. The output layers output the corresponding structural response to a linear mapping function as the implied layer, and the output layers complete the optimization calculation process.

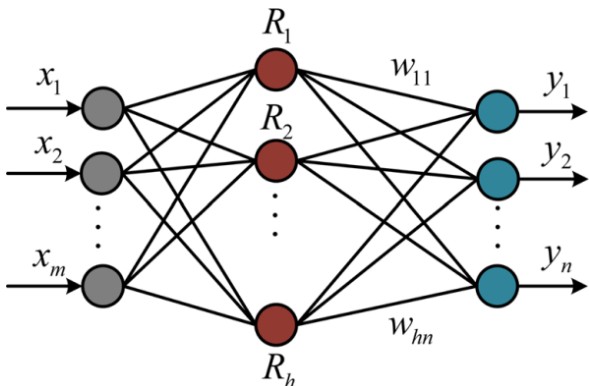

**Figure 1.** Radial basis neural network structure diagram.

The most distinctive feature of the RBF method is that its implicit layer nodes use the Euclidean distance function as computational input, converting a multidimensional optimization problem into a one-dimensional optimization problem with distance as a variable. The RBF method uses the radial basis function as the transfer function and maps the radial distance *r* from the input sample *x* to the center of the implicit layer as a variable with the following specific mathematical expression.

$$
\begin{cases}
f'(x) = \sum\limits_{i=1}^{n} w_{mk} v_k(x) + b_m \\
v_k(x) = \|x - x_k\|
\end{cases}
\tag{20}
$$

where: $f'(x)$ is an approximation of the true response value $f(x)$; $x = [x_1, x_2 \cdots \cdots x_n]$ is a *n*-dimensional inputs vector; *n* is the number of basis functions; *m* is the number of output responses; $w_{mk}$ is the connection weight between the *k*th implied layer node and the *m*th output layers node; $b_m$ is the deviation of the *m*th output response; $v_k(x)$ is a radial basis function indicating the distance between the sample *x* and the *i*th sample $x_i$ in the design space.

### 2.6. Optimize the Design Process

Figure 2 shows the main process and content of this study.

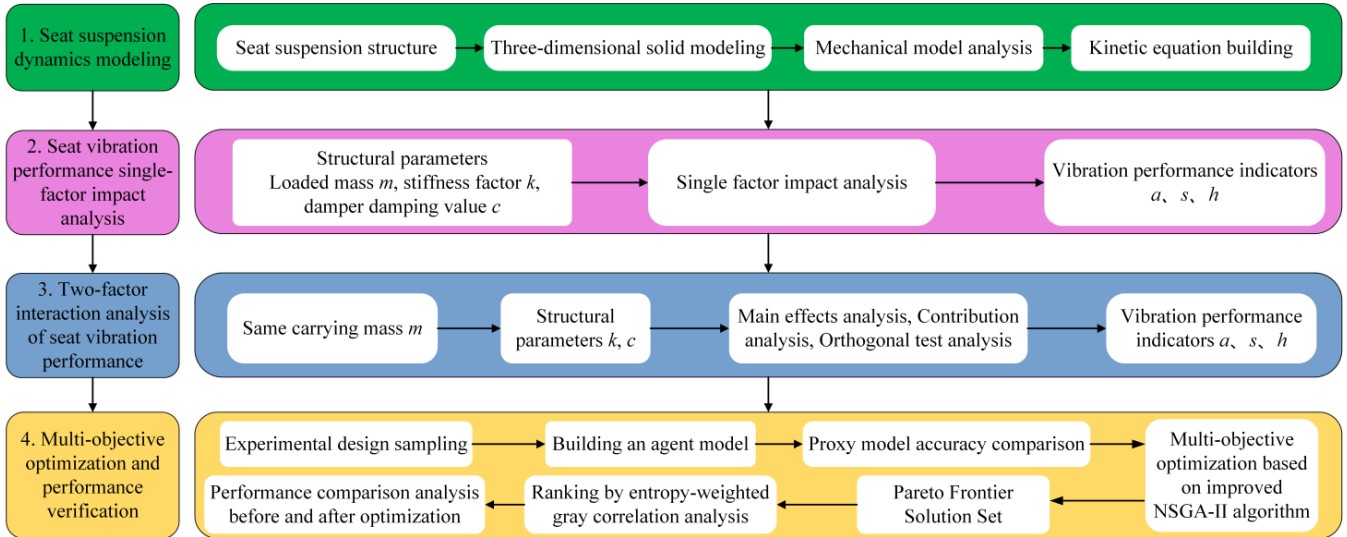

**Figure 2.** Flow chart of seat suspension vibration attenuation optimization design.

### 3. Scissor Seat Suspension Dynamics Model

#### 3.1. Seat Suspension Structure Form

The scissor seat suspension application tractor is shown in Figure 3a; the physical object is shown in Figure 3b–d. The 3D model is shown in Figure 3e, and the structural sketch is shown in Figure 3f. The seat suspension consists of upper and lower frames, scissor bars, springs, shock absorbers, and other components. Mutual articulation point E connects the seat suspension scissor rods. The scissor rod 3 is articulated with the seat's lower frame of point B, and its upper-end D point can slide into the upper frame's linear slots. The shear rod 4 articulates with the upper frame of the seat on pointing A, and its lower end C can slide into the linear slots of the seat's lower frame. Spring 2 is a linear coil spring with stiffness $k$. Spring 2 is placed horizontally in the upper frame. Spring 2 is connected to point G at the upper end of the scissor rod 4, and the other end is connected to the right side of the upper frame of the seat suspension. Damper 6 is located between the upper and lower frames of the seat suspension and has a damping factor of $c$. The shear bar has the dimensions $BD = l_1 + l_2$, $AC = l_3 + l_4$, $EA = l_4$, $EB = l_1 = EC = l_3$, $ED = l_2$. The angle between the shear bar and the bottom is $\phi$.

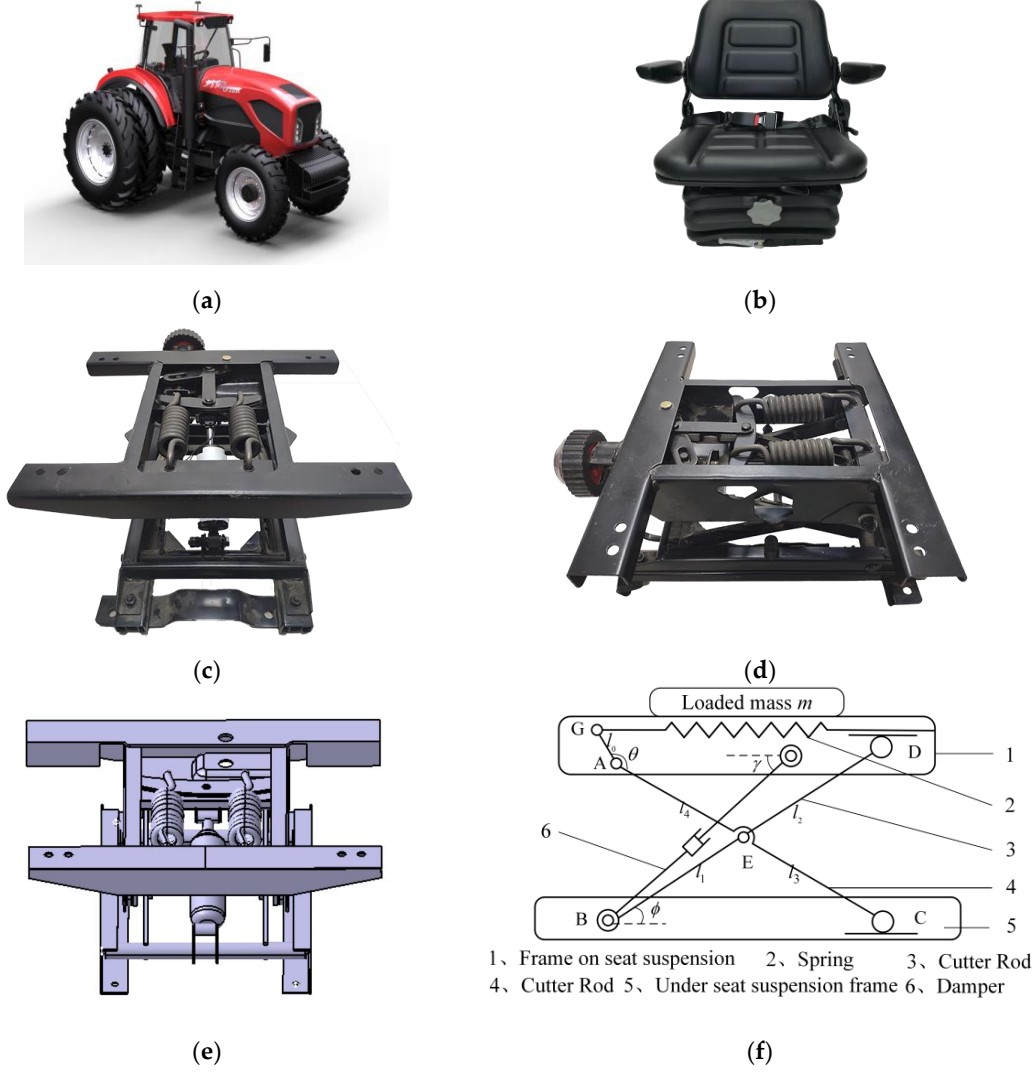

(a)　　　　　　　　　　　　(b)

(c)　　　　　　　　　　　　(d)

1、Frame on seat suspension　　2、Spring　　3、Cutter Rod
4、Cutter Rod　5、Under seat suspension frame　6、Damper

(e)　　　　　　　　　　　　(f)

**Figure 3.** Structure diagram of a type of scissor seat suspension. (**a**) Seat suspension for tractors. (**b**) Seat suspension shall be applied to the seat. (**c**) Picture of seat suspension (front view). (**d**) Picture of seat suspension (left view). (**e**) 3D model diagram of seat suspension. (**f**) Structure diagram of seat suspension.

### 3.2. Seat Suspension Mechanics Model

The following assumptions are made to simplify the equation's calculation: ① Regardless of the influence of the seat cushion between the human body and the upper plate of the seat suspension, 75% of the human body mass and the upper plate quality of the seat suspension is called the bearing mass $m$. ② In simplifying the equation, assume that the scissor rods have no mass. ③ The seat under excitation produces vibrations in a static equilibrium position.

The scissor rod motion is shown in Figure 4 under excitation $y_1$, with the seat base plate playing the role of displacement excitation $y_1 = Q \sin \omega t$, where $Q$ is the excitation amplitude, and $\omega$ is the excitation circle frequency. The upper panel of the seat's response are $y_2$, The relationship between spring deformation $\Delta_x$, $y_1$ and $y_2$ is then as shown in Equation (21):

$$\Delta_x = \frac{l_0 \{\cos[\theta' - (\phi + \Delta\phi)] - \cos(\theta - \phi)\}}{(l_3 + l_4)[\sin(\phi + \Delta\phi) - \sin\phi]}(y_2 - y_1) \tag{21}$$

where $\phi$ is the angle in static equilibrium between the shear bar and the horizontal direction of the seat floor, and $\Delta\phi$ is the change in angle under excitation.

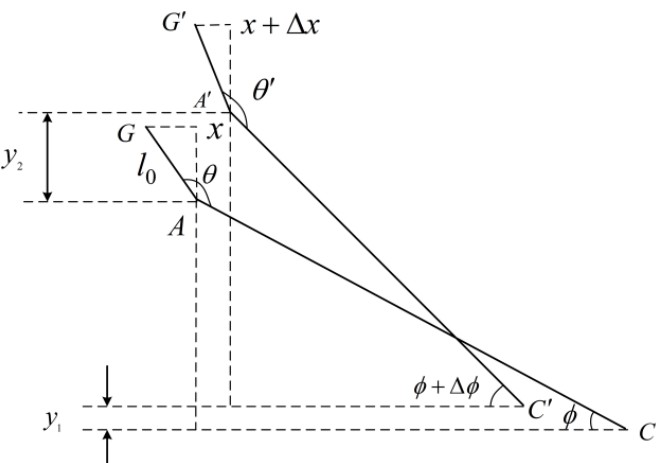

**Figure 4.** Shear $y_1$ bar motion under excitation.

Figure 5 depicts the overall structural force analysis of the seat suspension:

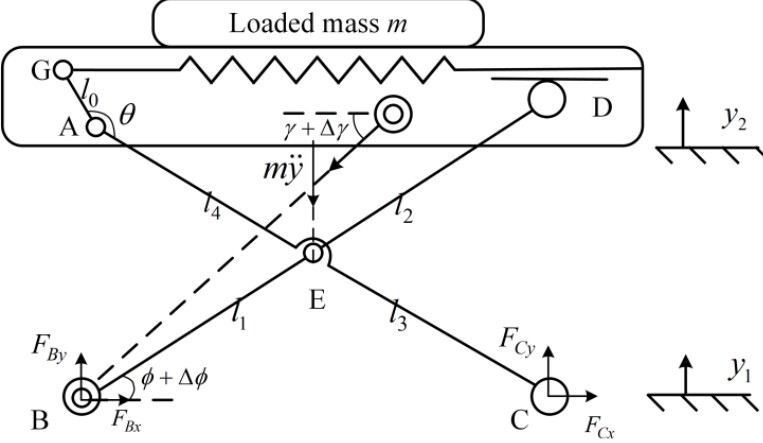

**Figure 5.** Overall force analysis of the seat.

The inertial force on the upper plate of the seat suspension during the vibration is $m\ddot{y}$, and the equilibrium equation can be established using D'Alembert's principle:

$$\begin{cases} F_{Bx} + F_{Cx} - F_d \cos(\gamma + \Delta\gamma) = 0 \\ F_{By} + F_{Cy} - m\ddot{y} - F_d \sin(\gamma + \Delta\gamma) = 0 \\ 2F_{Cy}l_1 \cos(\phi + \Delta\phi) - m\ddot{y}_2 l_1 \cos(\phi + \Delta\phi) = 0 \\ F_d = c(\dot{y}_2 - \dot{y}_1) \sin(\phi + \Delta\phi) \\ F_k = -k\Delta x \\ F_{Cx} = f_d F_{Cy} \end{cases} \tag{22}$$

where $F_d$ is the damper's damping force (N), $\gamma$ is the angle between the axial direction of the damper and the horizontal direction of the upper plate of the suspension in static equilibrium (°). $\Delta\gamma$ is the change in damper tilt due to the excitation effect. $f_d$ is the sliding friction factor. $F_k$ is the spring force (N).

Figure 6 depicts the shear rod force analysis, and the equilibrium equations of the shear rods AC and BD can be established.

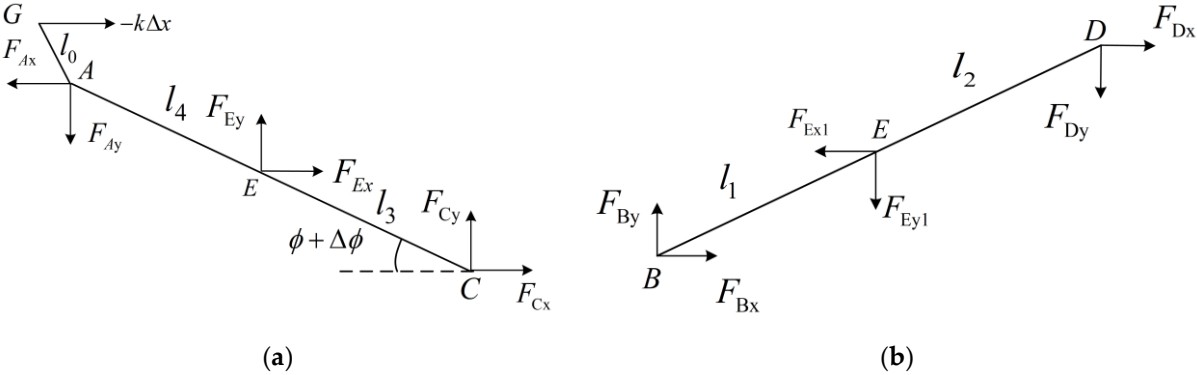

**(a)**                                       **(b)**

**Figure 6.** Shear bar force analysis diagram. (**a**) AC rod force analysis. (**b**) BD rod force analysis.

Force analysis of shear bar AC (see Figure 6a):

$$\begin{cases} F_{Cx} + F_{Ex} - F_{Ax} - k\Delta x = 0 \\ F_{Cy} + F_{Ey} - F_{Ay} = 0 \\ F_{Ax}l_4 \sin(\phi + \Delta\phi) + F_{Ay}l_4 \cos(\phi + \Delta\phi) + \\ F_{Cy}l_3 \cos(\phi + \Delta\phi) + F_{Cx}l_3 \sin(\phi + \Delta\phi) + \\ k\Delta x[l_4 \sin(\phi + \Delta\phi) + l_0 \sin(\theta - \phi - \Delta\phi)] = 0 \end{cases} \tag{23}$$

Force analysis of shear bar BD (see Figure 6b):

$$\begin{cases} F_{Bx} + F_{Dx} - F_{Ex1} = 0 \\ F_{By} - F_{Ey1} - F_{Dy} = 0 \\ F_{Bx}l_1 \sin(\phi + \Delta\phi) - F_{By}l_1 \cos(\phi + \Delta\phi) - \\ F_{Dx}l_2 \sin(\phi + \Delta\phi) - F_{Dy}l_2 \cos(\phi + \Delta\phi) = 0 \end{cases} \tag{24}$$

When $F_{Ex} = F_{Ex1}$, $F_{Ey} = F_{Ey1}$, the Equations (23) and (24) is solved by:

$$\begin{aligned} F_{Ex} = {} & \frac{l_{1,2}F_{Bx}}{2l_2} - \frac{l_{1,2}\cot(\phi+\Delta\phi)F_{By}}{2l_2} - \frac{l_{3,4}F_{Cx}}{2l_4} - \frac{l_{3,4}\cot(\phi+\Delta\phi)F_{Cy}}{2l_4} \\ & - \frac{k\Delta x l_0 \sin(\theta-\phi-\Delta\phi)}{2l_4 \sin(\phi+\Delta\phi)} \end{aligned} \tag{25}$$

$$\begin{aligned} F_{Ey} = {} & \frac{l_{1,2}F_{By}}{2l_2} - \frac{l_{1,2}\tan(\phi+\Delta\phi)F_{Bx}}{2l_2} - \frac{l_{3,4}\tan(\phi+\Delta\phi)F_{Cx}}{2l_4} - \frac{l_{3,4}F_{Cy}}{2l_4} \\ & - \frac{k\Delta x l_0 \sin(\theta-\phi-\Delta\phi)}{2l_4 \cos(\phi+\Delta\phi)} \end{aligned} \tag{26}$$

From $F_{Dx} = F_{Dy} f_d$ and Equation (23) we have:

$$F_{Bx} - F_{Ex} + (F_{By} - F_{Ey}) f_d = 0 \tag{27}$$

According to assumption ③, $\Delta\phi$ and $\Delta\gamma$ approaches zero, then $\phi + \Delta\phi \approx \phi$, $\gamma + \Delta\gamma \approx \gamma$. Substitute $F_{Ex}$, $F_{Ey}$ into Equation (27) to combine the above equations, and organize and simplify to obtain:

$$x_1 F_{Bx} + x_2 F_{By} + x_3 F_{Cx} + x_4 F_{Cy} + \frac{k\Delta x l_0 \sin(\theta - \phi)}{2l_4 \sin\phi} + \frac{k\Delta x l_0 \sin(\theta - \phi)}{2l_4 \cos\phi} f_d = 0 \tag{28}$$

In the formula, $x_1 = \left(1 - \frac{l_{1,2}}{2l_1} + \frac{l_{1,2}}{2l_2}\tan\phi f_d\right)$, $x_2 = \left(\frac{l_{1,2}}{2l_1}\cot\phi - \frac{l_{1,2}}{2l_2}f_d + f_d\right)$, $x_3 = \left(\frac{l_{3,4}}{2l_4} + \frac{l_{3,4}}{2l_4}\tan\phi f_d\right)$, $x_4 = \left(\frac{l_{3,4}}{2l_4}\cot\phi + \frac{l_{3,4}}{2l_4}f_d\right)$

Simplifying the collation yields:

$$m\ddot{y}_2 + A_1\dot{y}_2 + A_2 y_2 = A_1\dot{y}_1 + A_2 y_1 \tag{29}$$

In the formula, $A_1 = \frac{c[(l_2 - l_1)(\sin 2\gamma + 2f_d \sin^2\gamma)\tan\phi + l_{1,2}(2\sin 2\gamma + f_d \sin 2\gamma \tan^2\phi)]}{[2l_{1,2}(f_d \tan\phi + 1)]}$, $A_2 = \frac{k l_0^2}{l_{3,4}^2}(\sin\theta - \cos\theta \tan\phi)^2$.

The coefficients of Equation (28) are all constants derived from Equation (29), and the differential equation of motion of the seat are a linear constant coefficient differential equation. $A_1$ and $A_2$ represent the equivalent damping and equivalent stiffness of the seat suspension system, respectively, and their magnitudes are related to the seat's structural parameters.

## 4. Seat Suspension Vibration Performance Impact Analysis

### 4.1. Main Performance Parameters of the Seat Suspension

According to the literature, the range of load-bearing masses acting on the seat suspension system is $m$ (45 < $m$ < 85) kg [43]. In order to study the effect of different masses on the vibration transfer characteristics of the seat suspension, the vibration transfer function study is carried out with a minimum value of 45 kg, a middle value of 70 kg and a maximum value of 85 kg as the representatives of the load-bearing mass. A certain type of scissor seat structure produced by a company is used as the research object, and the static equilibrium position of the seat when 70 kg acts on the seat suspension system is used as the initial position of the seat suspension system.

### 4.1.1. Stiffness Coefficient $k$

The inherent frequency of the seat vibration system $f_0$ depends on the spring stiffness $k$, the load mass $m$, the angle between the shear bar and the horizontal direction of the seat base of static equilibrium $\phi$, and the geometric parameters of the shear bar, $\theta$, $\beta$, $l_0$ and $l_{3,4}$.

From Equation (29), the seat vibration damping inherent circular frequency $\omega_0$ and inherent frequency $f_0$ can be obtained as:

$$\omega_0 = \sqrt{\frac{A_2}{m}} = \frac{l_0}{l_{3,4}}(\sin\theta - \cos\theta \tan\phi)\sqrt{\frac{k}{m}} \tag{30}$$

$$f_0 = \frac{\omega_0}{2\pi} = \frac{l_0}{2\pi l_{3,4}}(\sin\theta - \cos\theta \tan\phi)\sqrt{\frac{k}{m}} \tag{31}$$

The inherent frequency $f_0$ of the seat suspension system should be (1 < $f_0$ < 2 Hz) [44], and the value of the seat suspension springs stiffness coefficient $k$ is (68.36 < $k$ < 273.44) N/mm when the bearing mass of the seat suspension is 70 kg, as obtained from Equation (31).

### 4.1.2. Damping Ratio $\zeta$

The damping ratio $\zeta$ to seat vibration depends on the damping coefficient of the damper $c$, the stiffness of the horizontal linear spring $k$, the load mass $m$, the inclination angle $\gamma$ of the damper at the static equilibrium position and the compression angle $\phi$ of the suspension, the sliding friction factor $f_d$ and the geometric parameters $l_0$, $l_1$, $l_2$, $l_{1,2}$ and $\theta$ of the seat suspension.

From Equation (29), the damping ratio when the seat is vibrating is:

$$\zeta = \frac{A_1}{2\sqrt{A_2 m}} = \frac{c[(l_2 - l_1)(\sin 2\gamma + 2f_d \sin^2 \gamma)\tan \phi + l_{1,2}(2\sin^2 \gamma + f_d \sin 2\gamma \tan^2 \phi)]}{[4l_0(f_d \tan \phi + 1)(\sin \theta - \cos \theta \tan \phi)\sqrt{mk}]} \tag{32}$$

The damper damping ratio $\zeta$ of the seat suspension system should be in the range of ($0.18 < \zeta < 0.35$) [44]. From Equation (32), when the seat suspension load mass is 70 kg, the seat suspension damper damping coefficient $c$ takes the value range of ($1.4 < c < 5.5$) N·s/mm.

### 4.1.3. Displacement $s$

Maximum displacement of the seat response surface is one of the performance evaluation parameters of seat suspension. According to the requirements of human comfort, the smaller the maximum displacement of the response surface, the better the vibration absorption performance. Let the action on the system excitation unit amplitude $y_1 = e^{i\omega t}$, and the response $y_2 = H(\omega)e^{i\omega t}$, the two equations will be derived and brought into the Formula (29) to obtain, eliminating $e^{i\omega t}$, the complex frequency response function is:

$$H(\omega) = \frac{\omega_0{}^2 + i2\zeta\omega_0\omega}{\omega_0{}^2 - \omega^2 + i2\zeta\omega_0\omega} \tag{33}$$

From Equation (33), the modal value of the complex frequency response function is:

$$|H(\omega)| = \sqrt{\frac{1 + (2\zeta\lambda)^2}{(1 - \lambda^2)^2 + (2\zeta\lambda)^2}}$$

By:

$$y_1 = Q \sin \omega t \tag{34}$$

The displacement function of the plane vibration on the seat is obtained as:

$$y_2(t) = Q\sqrt{\frac{1 + (2\zeta\lambda)^2}{(1 - \lambda^2)^2 + (2\zeta\lambda)^2}} \sin(\omega t - \varphi) \tag{35}$$

where $\lambda$ is the frequency ratio, $\lambda = \omega/\omega_0$; $\varphi$ is the phase difference angle between excitation and response. From Equations (32) and (35), the spring stiffness, $k$, and the damper damping, $c$, have a large effect on the magnitude of the performance evaluation parameter displacement, $s$.

### 4.1.4. Acceleration $a$

Performance evaluation parameters include the maximum acceleration of the plane response on the seat, and the smaller the maximum acceleration of the plane response on the seat, the better the vibration absorption performance. The acceleration calculation formula is shown in Equation (36).

The seat upper planes vibration response displacement from Equation (35) is $y_2(t)$, and the seat upper planes response acceleration $a = \ddot{y}_2(t)$:

$$a = -Q\omega^2 \sqrt{\frac{1 + (2\zeta\lambda)^2}{(1 - \lambda^2)^2 + (2\zeta\lambda)^2}} \sin(\omega t - \varphi) \tag{36}$$

From Equations (32) and (36), it can be seen that in the seat suspension system, the spring stiffness, $k$, and the damper damping, $c$, have a large effect on the magnitude of the performance evaluation parameter, acceleration, $a$.

4.1.5. Transmission Rate $\eta$

Performance evaluation parameters include the seat vibration transmission rate; the smaller the seat vibration transmission rate, the better the vibration attenuation. From Equations (34) and (35), the transmission efficiency calculation formula Equation (37) is obtained.

The vibration transmittance $\eta$ of the seat is the ratio of the response amplitude $Q\sqrt{\frac{1+(2\zeta\lambda)^2}{(1-\lambda^2)^2+(2\zeta\lambda)^2}}$ to the excitation amplitude $Q$, i.e.,:

$$\eta = \sqrt{\frac{1 + (2\zeta\lambda)^2}{(1 - \lambda^2)^2 + (2\zeta\lambda)^2}} \tag{37}$$

From Equations (32) and (37), it can be seen that in the seat suspension system, the spring stiffness, $k$, and the damper damping, $c$, have a large influence on the magnitude of the performance evaluation parameters $\eta$.

*4.2. Load Quality Impact Analysis*

The parameters of the seat of the initial position are shown in Table 1. The sliding friction factor $f_d$ is taken as 0.3. The initial position of the seat is ensured as a static equilibrium position in different suspension loads masses by the preload of the coil springs of the seat suspension.

**Table 1.** Seat suspension static balance position structure parameters.

| $l_0$/mm | $l_1$/mm | $l_2$/mm | $l_3$/mm | $l_4$/mm | $\theta$/(°) | $\gamma$/(°) | $\phi$/(°) |
|---|---|---|---|---|---|---|---|
| 50 | 132 | 107 | 132 | 120 | 115 | 20.6 | 11.6 |

In order to make $f_0$, $\zeta$ within a reasonable range of values, the values of seat suspension spring stiffness coefficient $k$ and damper damping coefficient $c$ at the equilibrium position are 172.28 N/mm and 3.37 N·s/mm, respectively. When the excitation amplitude is 2 mm, the input frequency is $f$ (0 < $f$ < 10) Hz, and the seat suspension carrying mass is 45 kg, 70 kg, and 85 kg. Substituting into Equations (31), (32), and (37), the seat vibration transmission characteristics under different seat suspension masses $m$ can be obtained, as shown in Figure 7.

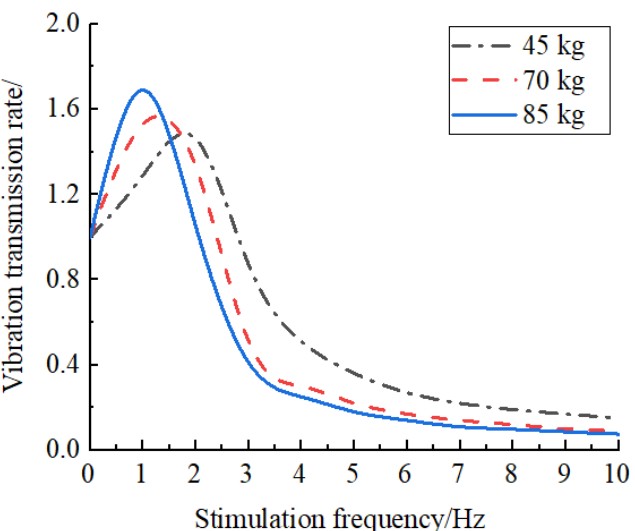

**Figure 7.** Vibration transmission characteristics under different seat suspension load mass.

The seat suspension vibration performance parameters for different suspension masses are shown in Table 2.

**Table 2.** Vibration performance parameters under different seat suspension load quality.

| Load Capacity of Different Suspensions/kg | Vibration Transfer Function Peak Frequency/Hz | Vibration Transfer Function Peak Amplitude/mm | Inherent Frequency/Hz | Damping Ratio | Vibration Transmission Rate at 3 Hz |
| --- | --- | --- | --- | --- | --- |
| $m = 45$ | 1.78 | 2.98 | 1.98 | 0.35 | 0.87 |
| $m = 70$ | 1.28 | 3.14 | 1.58 | 0.27 | 0.51 |
| $m = 85$ | 1.00 | 3.38 | 1.44 | 0.25 | 0.41 |

Under the rule of satisfying the suspension inherent frequency $f_0(1 < f_0 < 2)$ Hz. The damping ratio $\zeta(0.18 < \zeta < 0.35)$ constraints. Figure 7 and Table 2, show that the greater the load-bearing mass of three different seat suspensions, the greater the vibration amplitude in the vibration amplification area and the more obvious the vibration attenuation of the seat in the vibration isolation area.

*4.3. Spring Stiffness Coefficient Influence Analysis*

In the vibration isolation zone, when the input sinusoidal excitation amplitude is 2 mm and the input frequency is the seat suspension inherent frequency corresponding to different stiffnesses with fixed damping ($c = 3.37$ Ns/mm), respectively. The vibration performance parameters with different stiffness factors are shown in Table 3. The seat response acceleration and displacement curves ($m = 70$ kg) are shown in Figure 8a,b.

**Table 3.** Vibration performance parameters at different spring stiffness factors.

| Different Stiffnesses $k$/(N/mm) | Maximum Response Acceleration/ (mm/s²) | Maximum Response Displacement/ mm | Vibration Transmission Rate | Seat Suspension Inherent Frequency/Hz |
| --- | --- | --- | --- | --- |
| $k = 70$ | 87.02 | 114.48 | 1.708 | 1.01 |
| $k = 170$ | 255.12 | 115.61 | 1.336 | 1.58 |
| $k = 272$ | 469.13 | 116.39 | 1.083 | 1.99 |

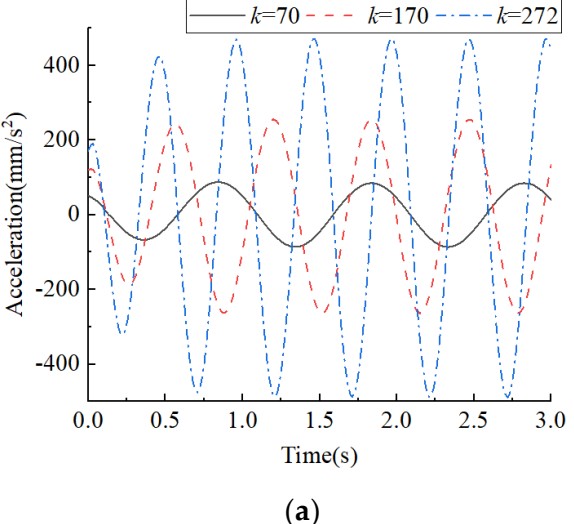
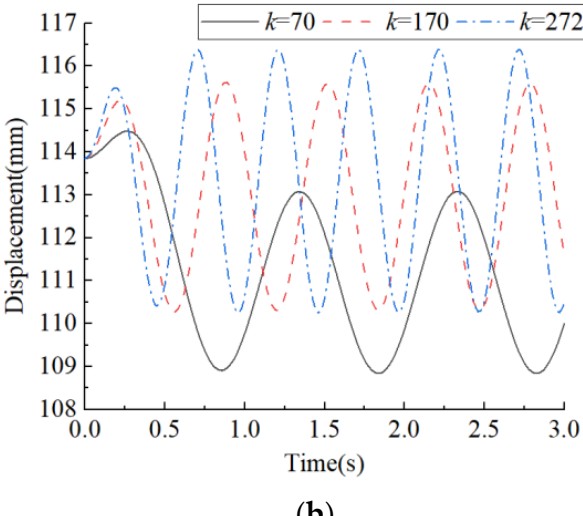

**(a)** **(b)**

**Figure 8.** Seat response acceleration curve with different spring stiffness values. (**a**) Acceleration Response. (**b**) Displacement Response.

When the load-bearing mass is 70 kg, with a reasonable range of values for the inherent frequency $f_0$ and damper damping ratio $\zeta$, the spring stiffness coefficient $k$ is obtained from Equation (31) in the range of (68.36 < $k$ < 273.44) N/mm. From Figure 8, it can be seen that in the range of $k$, with the increase of spring stiffness coefficient, the acceleration and displacement of the plane on the seat increase, and the vibration attenuation of the seat suspension decreases.

*4.4. Damper Damping Influence Analysis*

In the vibration isolation zone, when the input sinusoidal excitation amplitude is 2 mm and the input frequency is the seat suspension inherent frequency corresponding to different damping and fixed stiffness ($k$ = 172.28 N/mm), respectively. The vibration performance parameters with different damping coefficients are shown in Table 4. The seat response acceleration and displacement curves ($m$ = 70 kg) are shown in Figure 9a,b.

When the load-bearing mass is 70 kg, with a reasonable range of values for the inherent frequency $f_0$ and damper damping ratio $\zeta$, the damper damping coefficient $c$ is obtained from Equation (32) in the range of (1.4 < $c$ < 5.5) N·s/mm. From Figure 9, it can be seen that in the range of value $c$, with the increase of damping coefficient, the acceleration and displacement of the plane on the seat decrease, and the vibration attenuation of the seat suspension is improved.

**Table 4.** Vibration performance parameters under different damping coefficients of dampers.

| Different Damping $c$/(N·s/mm) | Maximum Response Acceleration/ (mm/s²) | Maximum Response Displacement/ mm | Vibration Transmission Rate | Seat Suspension Inherent Frequency/Hz |
|---|---|---|---|---|
| $c$ = 1.9 | 352.05 | 116.58 | 1.228 | 1.59 |
| $c$ = 3.7 | 248.36 | 115.52 | 1.043 | 1.59 |
| $c$ = 5.5 | 223.19 | 115.22 | 1.272 | 1.59 |

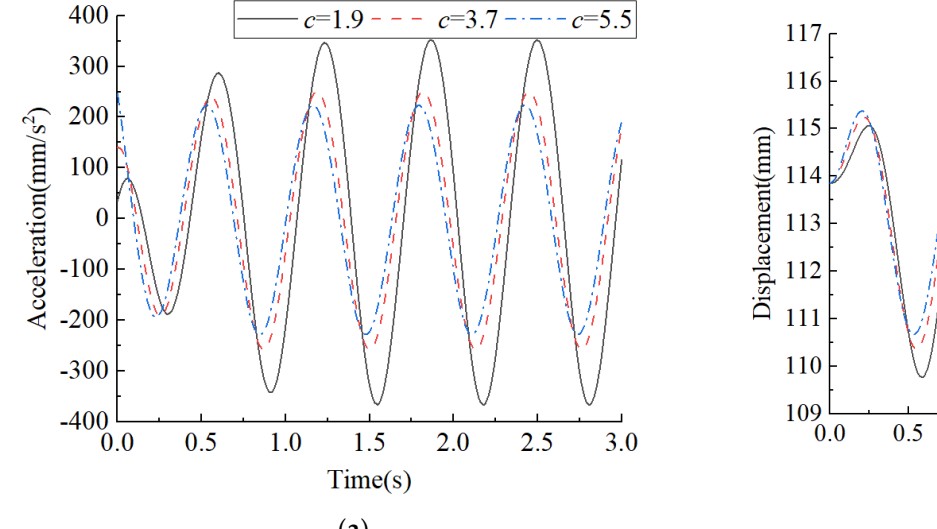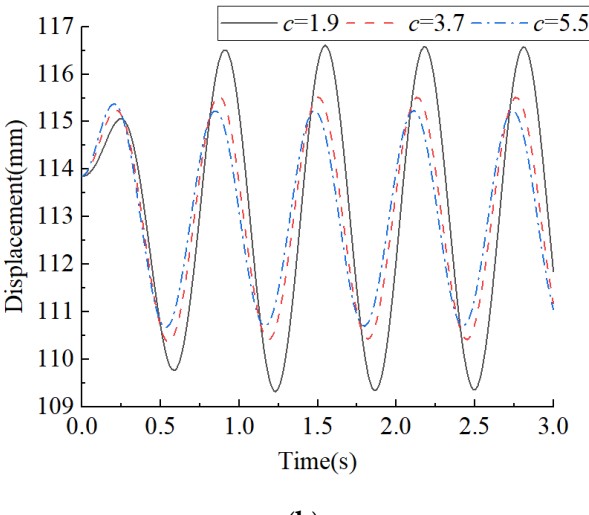

(**a**)                                    (**b**)

**Figure 9.** Acceleration and displacement curves of seat response with different damper damping values. (**a**) Acceleration Response. (**b**) Displacement Response.

*4.5. Analysis of Factors Influencing Different Performance Indicators*

(1)    Main Effect Analysis

According to the main effect analysis formula for Section 2.1, Table 5 shows the table of the main factor effect of the control factor when the single optimization target is in the upper plane of the seat, mainly listing the analysis results in the optimization target response acceleration $a$, response displacement $s$ and transfer rate $\eta$. Figure 10 shows the main effect plots of $a$, $s$, and $\eta$.

**Table 5.** Response results of each index under different variables.

| Main Effect Evaluation Indicators | | Acceleration $a$/(mm/s$^2$) | | Displacement $s$/mm | | Transmission Rate/ | |
|---|---|---|---|---|---|---|---|
| | | $k$ | $c$ | $k$ | $c$ | $k$ | $c$ |
| Level | 1 | 128.964 | 391.331 | 114.821 | 116.650 | 1.853 | 3.651 |
| | 2 | 182.668 | 235.396 | 115.262 | 115.451 | 1.902 | 2.294 |
| | 3 | 252.884 | 302.738 | 115.684 | 115.674 | 2.251 | 2.062 |
| | 4 | 335.443 | 215.925 | 116.062 | 115.322 | 2.420 | 1.671 |
| | 5 | 464.419 | 218.989 | 116.561 | 115.310 | 2.803 | 1.541 |
| The main Effect value | | 335.455 | 175.406 | 1.740 | 1.340 | 0.950 | 2.110 |
| Sorting | | 1 | 2 | 1 | 2 | 1 | 2 |
| Interaction effect value | | 132.672 | | 0.580 | | 0.705 | |
| The overall average | | 272.88 | | 115.68 | | 2.24 | |

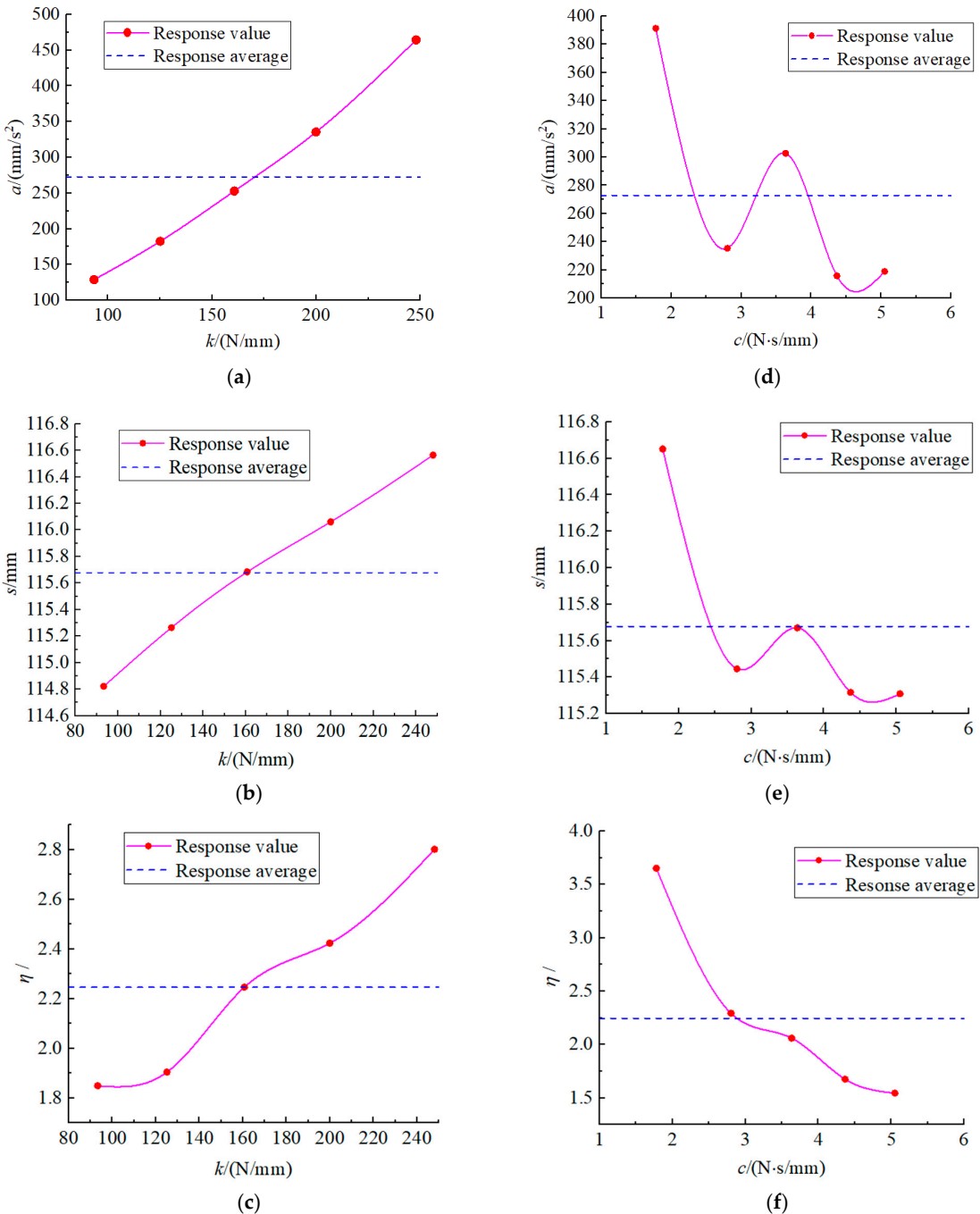

**Figure 10.** Main effect analysis diagram. (**a**) Influence of spring stiffness *k* on acceleration *a*. (**b**) Influence of spring stiffness *k* on displacement *s*. (**c**) Spring stiffness *k* to transmissibility *η* Impact analysis of. (**d**) Influence of damper damping *c* on acceleration *a*. (**e**) Influence of damper damping *c* on displacement *s*. (**f**) Damper damping *c* pair transmissibility *η* Impact of.

From Figure 10a,d, it can be seen that the main effect of acceleration *a* is that acceleration *a* increases with the increase of spring stiffness *k*, and damper damping *c* is non-linear to the fluctuation *a*. *k* plays a decisive role in *a*, followed by *c*. From Figure 10b,e, it can be seen that the main effect analysis regarding the displacement *s* yields that *k* becomes linearly and positively correlated with *s*. The effect of the damper *c* on *s* becomes nonlinear, with *k* having a decisive effect on *s* and *c* being the second most important. From Figure 10c,f, it can be seen that the effects of both *k* and *c* on the transfer rate *η* are volatile and become nonlinear. Therefore, the seat suspension system is a multi-factor interaction system. In

response to the fluctuation of the relationship between both influencing factors $k$ and $c$ on $\eta$, it is not clear that $k$ or $c$ has a decisive influence on $\eta$. Contribution analysis is needed to verify further the results of the main effect analysis in Figure 10.

(2)  Contribution analysis

According to the contribution degree analysis formula for Section 2.2, 80 sample points were designed and calculated by the Latin hypercube method, and the contribution degree analysis method was selected to calculate the acceleration $a$, displacement $s$, and transmission rate $\eta$ of the upper plane of the seat suspension as response indicators. The results of some sample points were listed as shown in Table 6.

**Table 6.** Results of experimental design of contribution.

| No. | Design Variables | | Performance Response | | |
|---|---|---|---|---|---|
| | $k$(N/mm) | $c$(N·s/mm) | $a$(mm/s$^2$) | $s$(mm) | $\eta$ |
| 1 | 95.055 | 4.394 | 119.380 | 114.930 | 1.449 |
| 2 | 261.120 | 3.403 | 442.620 | 116.310 | 2.457 |
| 3 | 264.008 | 3.403 | 442.620 | 116.310 | 2.457 |
| $\ldots$ | $\ldots$ | $\ldots$ | $\ldots$ | $\ldots$ | $\ldots$ |
| 79 | 73.990 | 4.694 | 104.120 | 114.810 | 1.323 |
| 80 | 258.415 | 1.648 | 712.660 | 118.200 | 4.718 |

From Figure 10, it can be seen that $k$ has a decisive effect on both $a$ and $s$; $c$ has the second most important effect on $a$ and $s$, consistent with the contribution analysis in Text 4.5(2). From Figure 11, $c$ has a decisive effect on the transfer rate $\eta$, and $k$ has the second most important effect on $\eta$.

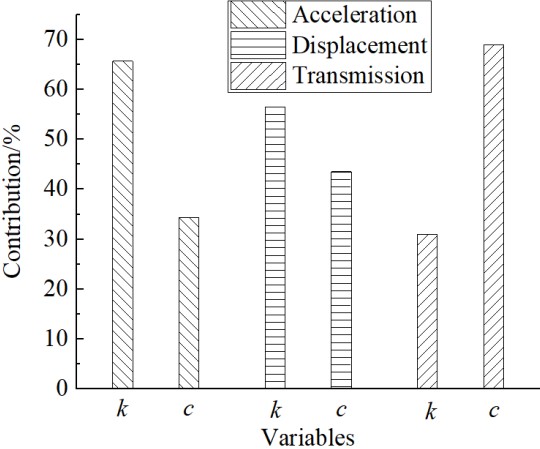

**Figure 11.** Analysis of the contribution of each indicator.

In summary, corresponding to the seat suspension of a certain bearing mass, the suspension system stiffness coefficient $k$ and damping $c$ determine the seat vibration performance and also affect the response indexes such as seat vibration acceleration $a$, vibration displacement $s$, and transmission rate $\eta$. The stiffness coefficient $k$ has a greater influence on the acceleration $a$ and vibration displacement $s$, and the damping $c$ has a greater influence on the transmission rate $\eta$. Response indices are interrelated and influenced, and one of the optimum may degrade other performance indices, resulting in a reduction in the overall seat vibration performance. Therefore, multi-objective optimization of seat vibration attenuation is required in order to obtain a compromise solution with excellent comprehensive performance.

## 5. Seat Suspension Vibration Attenuation Multi-Objective Optimization

### 5.1. Approximate Model Approach

The Optimal Latin Hypercube Sampling (OLHS) was used for the Design of Experiments (DoE) sampling to obtain experimental samples of different variable parameters (*k*, *c*). A total of 80 sample points were selected to fit the Kriging and RBF approximation models for each performance index.

In order to ensure the accuracy of multi-objective optimization, the accuracy of the approximation models needs to be verified. The higher the precision, the higher the confidence in the optimal solution. The coefficient $R^2$ is often used as an evaluation index in engineering. The closer the $R^2$ value is to 1, the higher the overall forecast accuracy of the approximate model is. Its mathematical expression is as follows.

$$R^2 = \frac{\sum\limits_{i=1}^{n} (\hat{y}_i - \overline{y})^2}{\sum\limits_{i=1}^{n} (y_i - \overline{y})^2} \tag{38}$$

where: $n$ is the number of test sample points; $\hat{y}_i$ and $y_i$ are the predicted and actual response values of the approximate model corresponding to the $i$th sample point, respectively; $\overline{y}$ is the average of the actual responses to all sample points.

Within the range of values of design variables, another 20 sample points were selected using the central composite design. The accuracy of the Kriging and RBF approximation models were verified. It used the cross-validation method. The accuracy verification results of the approximation model are shown in Figure 12a–d.

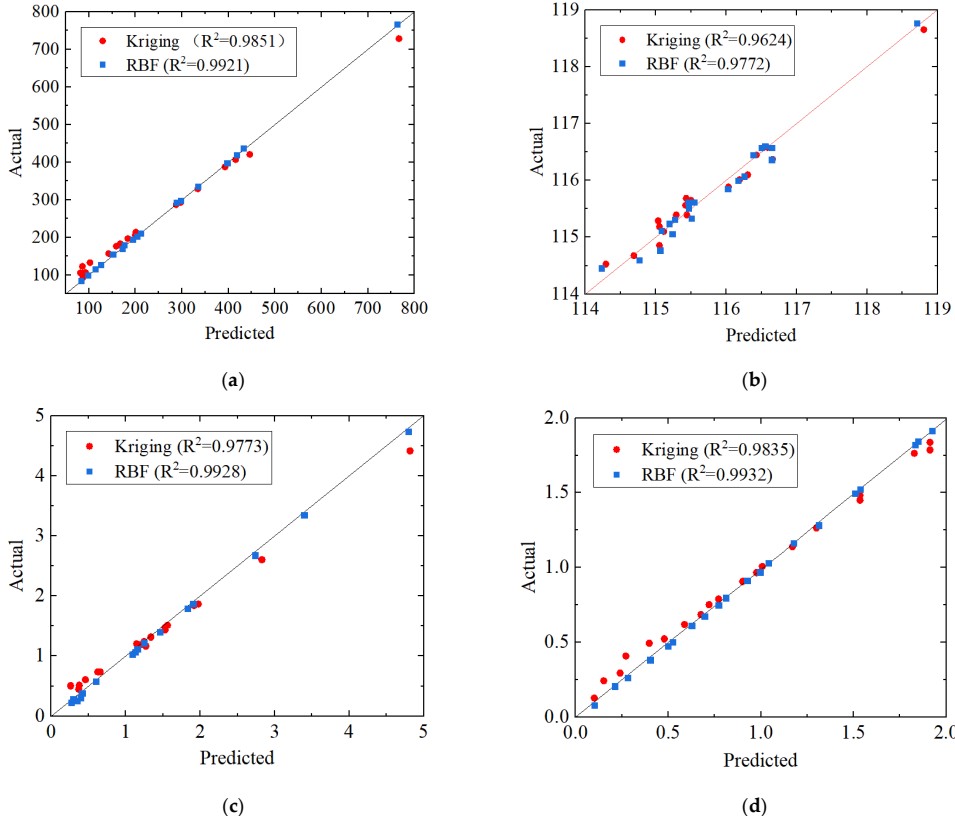

**Figure 12.** RBF and Kriging fitting model for vibration attenuation index. (**a**) Accuracy verification of acceleration *a* approximate model. (**b**) Accuracy verification of displacement *s* approximate model. (**c**) Transmissibility $\eta$ Accuracy verification of approximate model. (**d**) Accuracy verification of approximate model of natural frequency $f_0$.

As can be seen from Table 7, the value of the coefficient of determination $R^2$ of the RBF proxy model is larger than that of the Kriging proxy model. Therefore, the RBF prediction model has better accuracy, and the RBF approximation model is used as the proxy model for the initial samples in this paper.

**Table 7.** Proxy model accuracy evaluation index.

| Performance Response | Kriging Surrogate Model | RBF Surrogate Model |
|---|---|---|
| | $R^2$ | $R^2$ |
| $a/(\text{mm/s}^2)$ | 0.9851 | 0.9921 |
| $s/\text{mm}$ | 0.9624 | 0.9772 |
| $\eta/$ | 0.9773 | 0.9928 |
| $f_0/$ | 0.9835 | 0.9932 |

### 5.2. Multi-Objective Optimization Methods

#### 5.2.1. Improved NSGA-II Algorithm

The NSGA-II was proposed by K Deb after introducing an elite retention strategy, fast non-dominated sorting method, and crowded distance comparison method based on the NSGA algorithm [45]. NSGA-II algorithm avoids the shortcomings of the NSGA algorithm due to the lack of diversity in the population due to shared fitness and improves the optimization efficiency and computational accuracy.

Although the crowded distance comparison method can maintain population diversity better, it is still inadequate for optimization problems with more than two objective functions. In order to further improve the population diversity of multi-objective optimization problems, a fixed threshold $\varepsilon$ elimination strategy is used in MSGA-II to replace the crowding distance comparison method, which can solve the multi-objective optimization problems more reasonably. The principle of the MNSGA-II algorithm is shown in Figure 13.

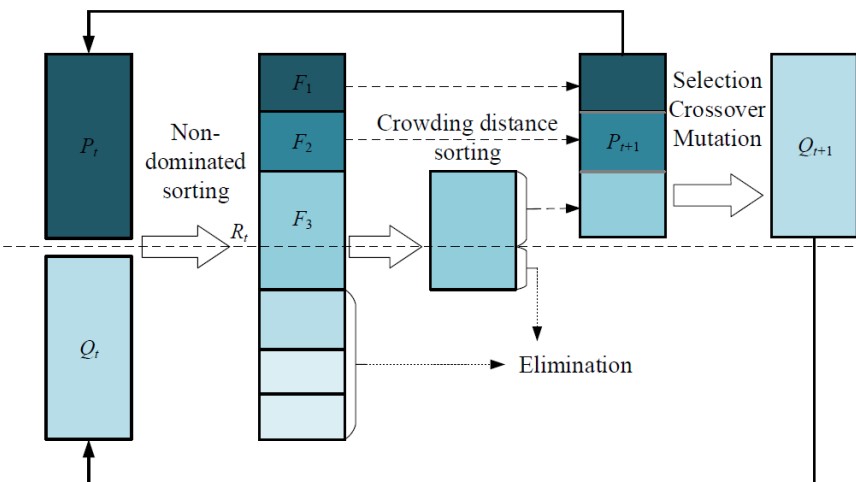

**Figure 13.** Principle of MNSGA-II algorithm.

The MNSGA-II algorithm operates as follows:

From Figure 14, we can see the operation steps of MNSGA-II algorithm.

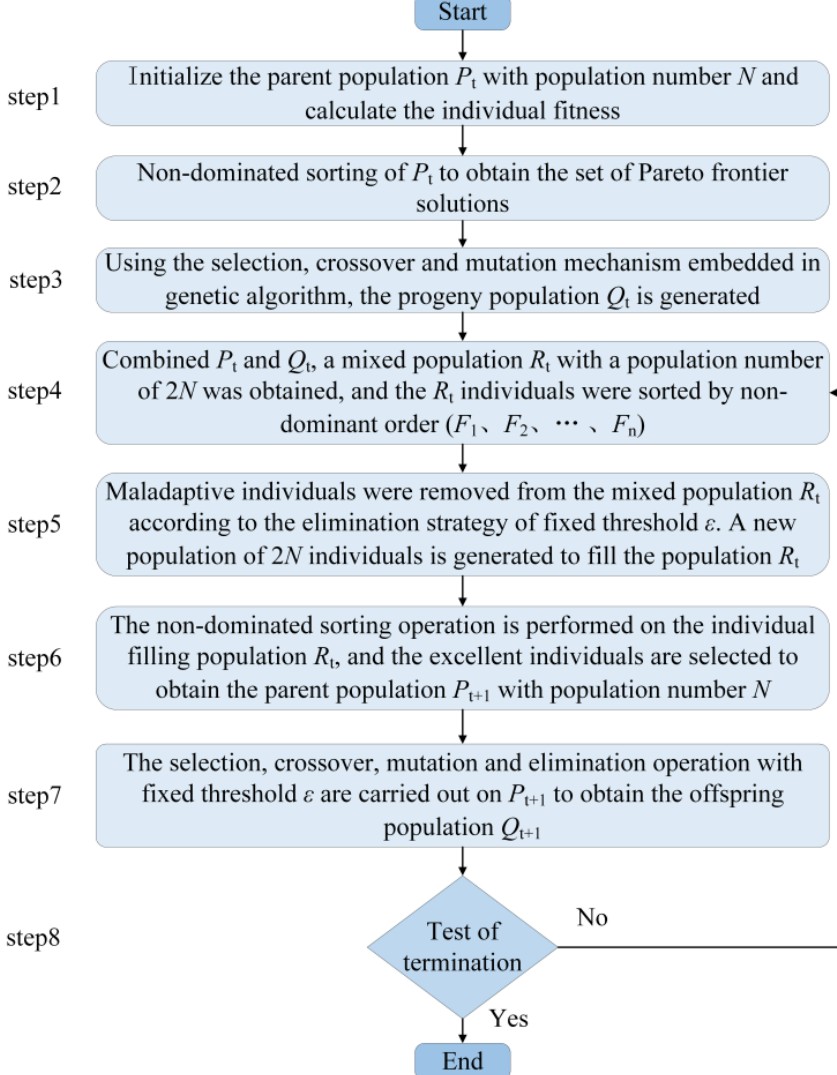

**Figure 14.** MNSGA-II algorithm step flow chart.

5.2.2. Multi-Objective Optimal Design

Based on the constructed RBF proxy model on the multi-objective optimization platform Isight. Multi-objective optimization is carried out by using an improved NSGA-II optimization algorithm. The optimized population size is 20, evolutionary algebra is 200, and crossover probability is 0.9. The Pareto solution is obtained by optimization through 4001 iterations. As shown in Figure 15a shows.

The green dots in Figure 15 show the compromise solutions recommended by the optimization platform after the multi-objective optimization. According to Figure 15a–d, it can be seen that the acceleration $a$, displacement $s$, and transfer rate $\eta$, the three objective values, are interrelated and affect each other; one optimal often brings another worse performance, so the three selections are a result of compromise. Therefore, it is necessary to find a method to balance the performance of the three and to select the solution with the better comprehensive performance of the Pareto solution set, and the green solution recommended by the Isight platform does not meet the value range of the seat suspension damping value $\zeta$, so it needs to be reselected. In this paper, the entropy-weighted gray correlation analysis ranking method is used to rank the comprehensive performance of Pareto solutions.

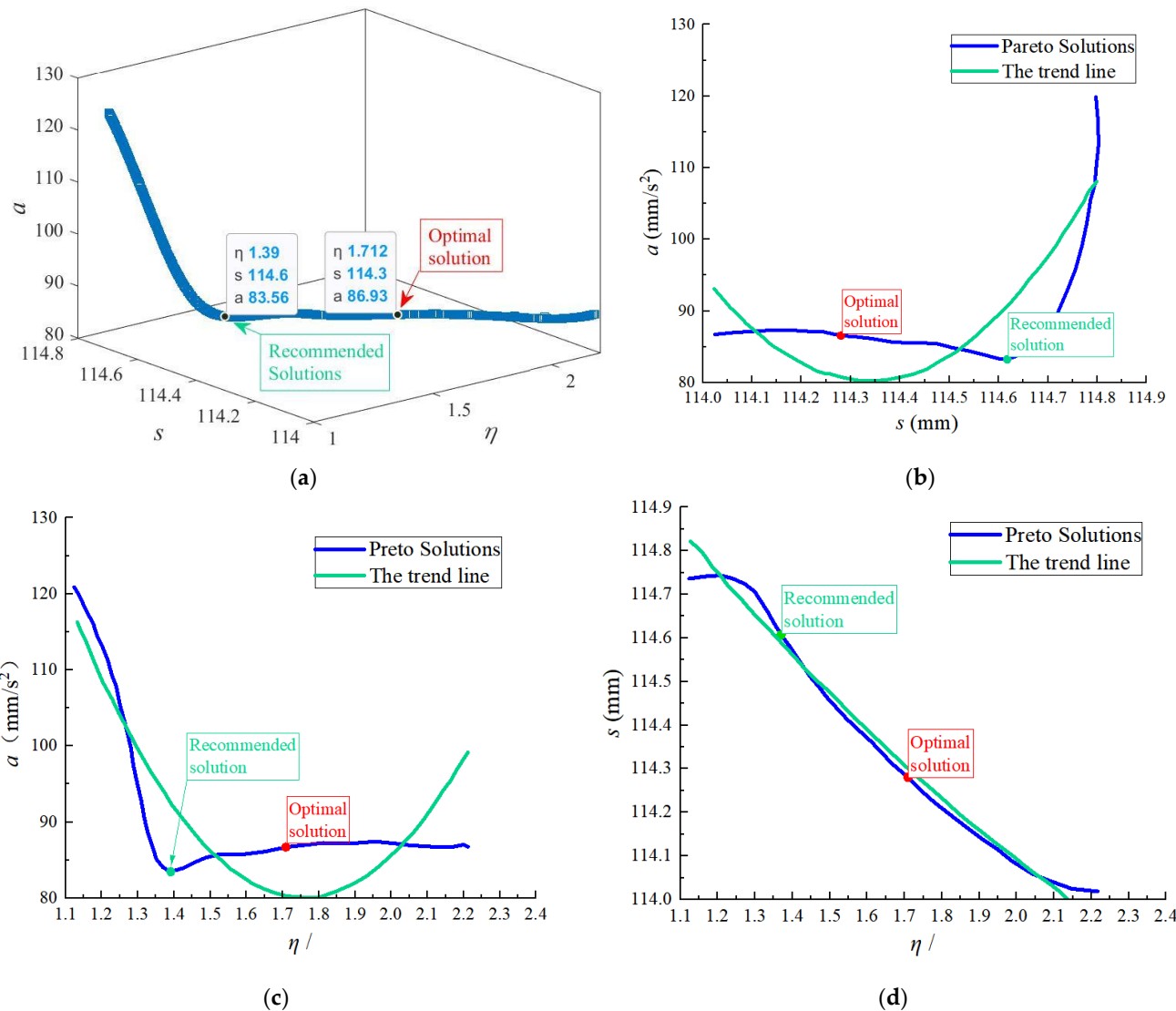

**Figure 15.** RBF optimization of each vibration attenuation index. (**a**) Preferred solutions for maximum acceleration, maximum displacement, and transfer rate. (**b**) Preferred solution for maximum acceleration and maximum displacement. (**c**) Preferred solution for maximum acceleration and transfer rate. (**d**) Preferred solution for maximum displacement and transfer rate.

*5.3. Entropy-Weighted Gray Correlation Ranking*

The three objective functions correspond to the 300 Pareto frontier solutions of Figure 15a: $a$, $s$, and $\eta$, all of which are required to be as small as possible. The objective functions are first normalized to derive the gray correlation coefficient, and then the gray correlation degree of each scheme is obtained based on the objective weight values. The gray correlation coefficient and gray correlation degree values of each Pareto solution are calculated according to the formulas in Sections 2.3 and 2.4. The results are shown in Table 8, which ranks the Pareto frontier solutions and shows the design scheme with the best overall performance. Figure 16 shows the gray correlation values of the Pareto frontier solutions. The gray correlation for the 203rd design solution has a maximum value of 0.928. The recommended solution corresponding to the maximum value of gray correlation is shown in Figure 15a.

**Table 8.** Gray correlation coefficients and gray correlation values.

| No. | Number of Gray Correlations for Each Response Index | | | Gray Correlation | Sort by |
|---|---|---|---|---|---|
| | $a$(mm/s$^2$) | $s$(mm) | $\eta$ | | |
| 1 | 0.516 | 0.667 | 0.632 | 0.589 | 179 |
| 2 | 0.524 | 0.560 | 0.692 | 0.568 | 186 |
| ... | ... | ... | ... | ... | ... |
| 165 | 0.926 | 0.887 | 0.763 | 0.882 | 2 |
| ... | ... | ... | ... | ... | ... |
| 203 | 0.967 | 0.896 | 0.876 | 0.928 | 1 |
| ... | ... | ... | ... | ... | ... |
| 300 | 0.763 | 0.853 | 0.634 | 0.764 | 82 |

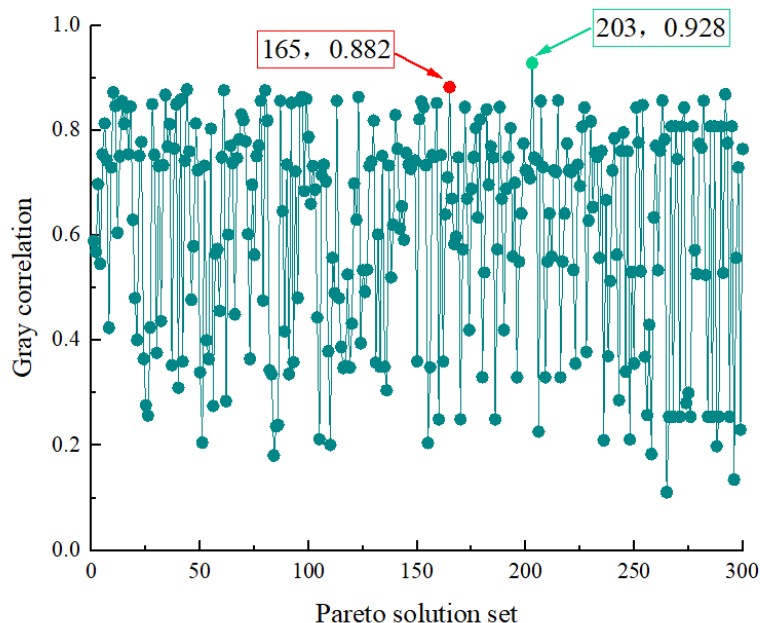

**Figure 16.** Gray correlation of the Pareto front for vibration attenuation.

The gray correlation degree value is based on Equation (16), and the GRG of each design solution can be obtained. Based on the magnitude of the GRG, each design solution can be ranked to obtain the best solution in terms of overall performance. The design solution to the highest GRG value is calculated to represent the solution to the best overall performance.

According to the gray correlation sorting, the sorting first is 203 group, the gray correlation value is 0.928, the corresponding $k$ and $c$ are substituted into the Formula (31) test, and the recommended solution is discarded because it does not meet the $\zeta$ taking value range. The second-ranked group is 165, and the gray correlation value is 0.882. At this time, $k$ = 68.36 N/mm and $c$ = 2.77 N·s/mm meet the requirements of the $\zeta$-taking range and are selected as the preferred solution for the optimization of this paper.

### 5.4. Comparison of Vibration Vibration Attenuation before and after Optimization

To verify the effectiveness of the multi-objective optimization, the combination of the Pareto preferred solution $k$ = 68.36 N/mm and $c$ = 2.77 N·s/mm was substituted into Adams with the initial $k$ = 172.28 N/mm and $c$ = 3.37 Ns/mm combination of the seat suspension for simulation analysis and comparative verification. The comparison of seat vibration attenuation parameters before and after optimization is shown in Table 9, and the results of response indexes ($a$, $s$, $\eta$) are shown in Figure 17.

**Table 9.** Comparison of seat vibration attenuation parameters before and after optimization.

| Vibration Performance Parameters | $k$/(N/mm), $c$/(N·s/mm) Before and After Optimization | | Performance Improvement Rate/% |
|---|---|---|---|
| | $k = 172.28$, $c = 3.37$ | $k = 68.36$, $c = 2.77$ | |
| Maximum response acceleration/(mm/s$^2$) | 259 | 87 | 66.41 |
| Maximum response displacement/(mm) | 115.64 | 112.97 | 2.31 |
| Rate of vibration transmission/ | 1.257 | 1.360 | 8.19 |
| Seat suspension inherent frequency/Hz | 1.56 | 1.00 | 35.90 |

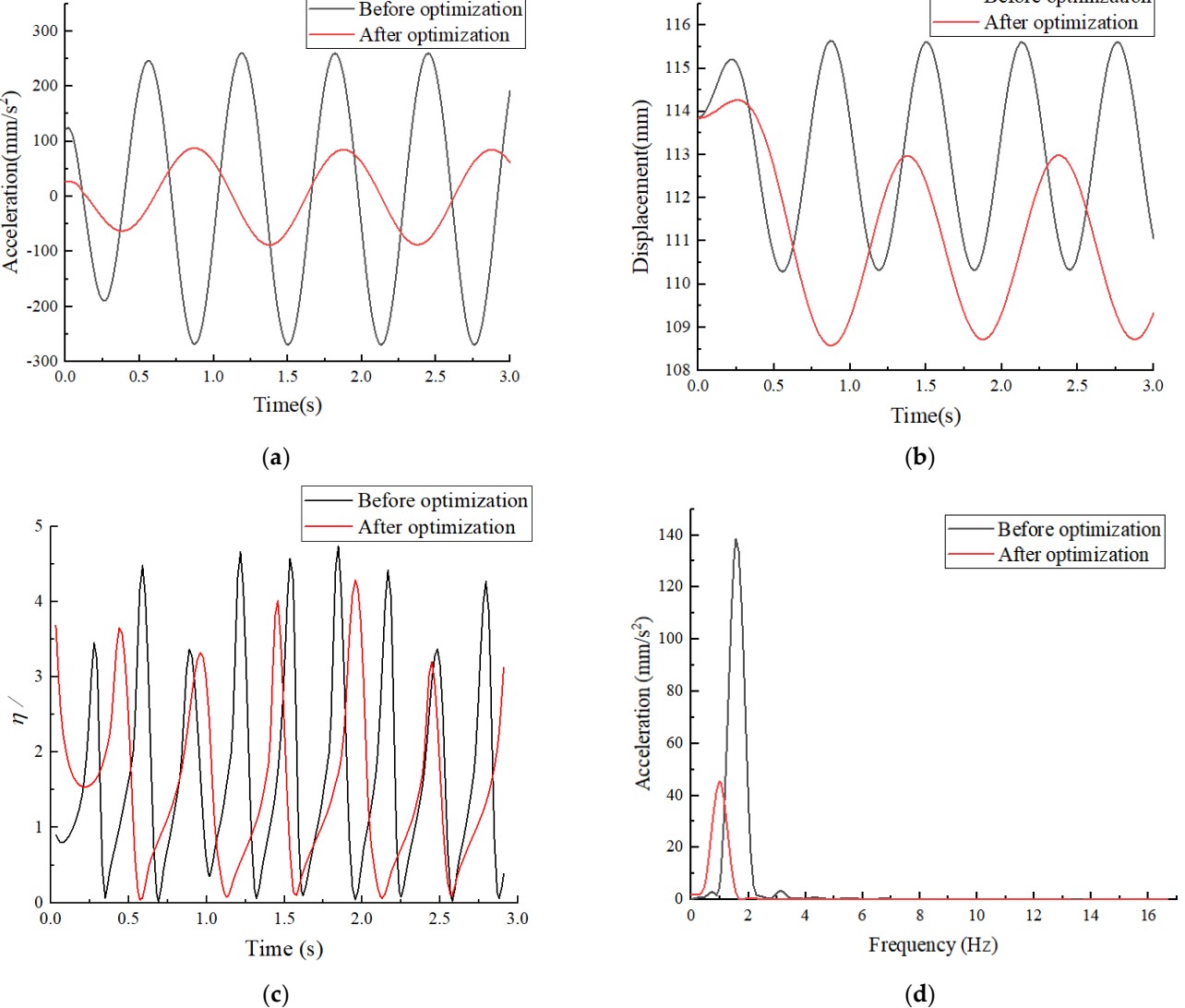

**Figure 17.** Comparison of each response index before and after optimization. (**a**) Comparison of acceleration before and after optimization. (**b**) Comparison of displacement before and after optimization. (**c**) Comparison of delivery rate before and after optimization. (**d**) Comparison of acceleration before and after optimization on frequency.

When the sinusoidal displacement excitation amplitude is 2 mm and the input frequency is the seat suspension natural frequency corresponding to different stiffness $k$ and damping $c$, respectively. The results are shown in Figure 17a–c, where the acceleration $a$ steady-state response value is significantly reduced, the displacement $s$ steady-state response value is reduced, and the transmittance $\eta$ is improved in general. The response

indexes *a*, *s*, and $\eta$ vibration attenuation are improved by 66.41%, 2.31%, and 8.19%, respectively, from Table 9.

## 6. Conclusions

This paper presents a multi-objective optimization design for seat suspension vibration attenuation based on an improved NSGA-II algorithm. A seat suspension dynamics model is established. The accuracy of the RBF approximation model is verified by the cross-validation method. Under different spring stiffness coefficients and damper damping coefficients, Pareto solutions are ordered by means of gray correlation analysis with entropic weight. The response index of plane vibration on the seat is analyzed, and the following conclusions are obtained:

(1) The equivalent stiffness and equivalent damping of the seat suspension are derived. The equivalent stiffness is mainly related to the spring stiffness, and the equivalent damping is mainly related to the damper damping. The scissor seat suspension under micro-amplitude vibration conditions can be approximated as a linear vibration system. In the range of values of seat suspension inherent frequency $f_0$ and damping ratio $\zeta$, the larger the seat suspension load mass, the larger the vibration amplitude of the seat on the vibration amplification zone and the greater the vibration attenuation in the vibration isolation zone. When the seat suspension load mass increased by 55.56% and 88.89%, the seat natural frequency decreased by 20.20% and 27.27%, the damping ratio decreased by 22.86% and 28.57%, and the seat vibration peak increased by 5.37% and 13.42%.

(2) Within the constraint range of seat suspension inherent frequency $f_0$ and damping ratio $\zeta$: under the condition of certain value of damper damping, as the value of spring stiffness increases, the value of acceleration and displacement of seat upper plane response increases, and the vibration attenuation ability of seat decreases. Under the condition of a certain value of spring stiffness, as the value of damper damping increases, the value of acceleration and displacement of seat upper plane response decreases, and the vibration attenuation of seat suspension improves.

(3) Through main effect analysis and contribution analysis, the relationship between the control variable and response index is obtained. The effect of *k* on the response to *a* and *s* is decisive, and the effect of *c* on the response from *a* and *s* is second. *c* on the response to $\eta$ is decisive, and the effect of *k* on the response of $\eta$ is second.

(4) RBF proxy model is constructed by combining experimental design sampling. Based on MNSGA-II multi-objective optimization algorithm, multi-objective optimization of seat suspension vibration performance is conducted, and Pareto frontier disaggregation is obtained. The entropy-weighted gray correlation ranking method is used to obtain the preferred solution that satisfies the range of variables. The effectiveness of the multi-objective optimization in this paper is verified by simulation analysis, and the response indexes *a*, *s*, and $\eta$ vibration attenuation are improved by 66.41%, 2.31%, and 8.19%, respectively, which effectively improves the seat suspension vibration attenuation. The method used in this paper provides a reference for the study of vibration attenuation control of seat suspension.

**Author Contributions:** Conceptualization, S.Z. and W.W.; methodology, S.Z. and W.W.; software, W.W.; validation, L.X., S.Z. and X.C.; formal analysis, Y.C.; investigation, X.C.; resources, S.Z.; data curation, W.W.; writing—original draft preparation, W.W.; writing—review and editing, W.W.; visualization, W.W.; supervision, L.X.; project administration, L.X.; funding acquisition, L.X. All authors have read and agreed to the published version of the manuscript.

**Funding:** This research was funded by the Major Science and Technology Project of Henan Province, grant number 221100240400, and by the Key Scientific Research Project of Colleges and Universities in Henan Province, grant number 20A460013.

**Institutional Review Board Statement:** Not applicable.

**Data Availability Statement:** Not applicable.

**Conflicts of Interest:** The authors declare no conflict of interest.

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
