# Peer review of "Vibration Performance Analysis and Multi-Objective Optimization Design of a Tractor Scissor Seat Suspension System"

_agriculture, doi:10.3390/agriculture13010048_

Round 1
Reviewer 1 Report
The article thoroughly analysis the influence of parameters on the vibration attenuation of seat suspension. The topic of the article is interesting, and extensive simulation and analysis have been made. The paper is suitable for publication in Agriculture. However, major revisions must be made before the final decision. The followings are my comment:
1) English writing should be largely improved. There are so many grammar mistakes. Don't write sentences that are too long to understand. Please check the attached file. I have marked most of the problems.
2) It is better to change 'damping performance to 'vibration control/attenuation performance' for a correct demonstration.
3) Abstract. It is not right to claim '... are improved by...-8.19%'. A negative improvement is here.
4) Introduction. Use 'et al.' when there are three or more authors of one paper.
5) The paper claims the application is for tractors and indicates 'there are few studies and applications on tractors.' What's the difference between tractors from other heavy-duty vehicles? Why are tractors so especially and need to be investigated individually? This information should be presented in the Introduction, and relevant backgrounds should be provided as well.
6) lines 470-472. The statement cannot be seen from Fig. 9.
7) Fig.10. I suggest the subfigures be arranged following a sequence of (a) (d); (b) (e);(c) (f) for a better presentation.
8) Fig. 14. Change the colour of green to dark green, as it is too bright.

Reviewer 2 Report
Please see the attachment for detailed comments.

Author Response
请参阅附件

Round 2
Reviewer 1 Report
The problem has been well addressed. However, the author may have over-corrected the paper towards the reviewer's comment: 'Don't write sentences that are too long to understand'.
For instance, a sentence: 'From the above study, it is clear that the stiffness and damping matching are the key factors to improve the damping performance of the seat suspension.' was revised to 'From the above study, it is clear that the stiffness and damping matching are the key factors. To improve the vibration attenuation of the seat suspension.', which is not correct in grammar. The author changed many long sentences which are correct to short sentences which are wrong in grammar. These revisions are found throughout the whole paper.
Technically, the paper can be accepted after all the grammar mistakes are revised. I suggest the author find a paper revision service to largely improve the quality of the paper in terms of grammar and expression.
